# Purported Self-Organized Criticality of the Cardiovascular Function: Methodological Considerations for Zipf’s Law Analysis

**DOI:** 10.3390/e26060496

**Published:** 2024-06-07

**Authors:** Jacques-Olivier Fortrat

**Affiliations:** 1Université d’Angers, CHU Angers, Inserm, CNRS, MITOVASC, Équipe CARME, SFR ICAT, 49000 Angers, France; jofortrat@chu-angers.fr; Tel.: +33-2-41-35-36-89; Fax: +33-2-41-35-50-42; 2Médecine Vasculaire, Centre Hospitalier Universitaire d’Angers, 4. rue Larrey, 49933 Angers Cedex 01, France

**Keywords:** baroreflex, blood pressure, heart rate variability, self-organized criticality, syncope, Zipf’s law

## Abstract

Self-organized criticality is a universal theory for dynamical systems that has recently been applied to the cardiovascular system. Precise methodological approaches are essential for understanding the dynamics of cardiovascular self-organized criticality. This study examines how the duration and quality of data recording affect the analysis of cardiovascular self-organized criticality, with a focus on the beat-by-beat heart rate variability time series obtained from seven healthy subjects in a standing position. Drawing a Zipf diagram, we evaluated the distribution of cardiovascular events of bradycardia and tachycardia. We identified tipping points for the distribution of both bradycardia and tachycardia events. By varying the recording durations (1, 2, 5, 10, 20, 30, and 40 min) and sampling frequencies (500, 250, and 100 Hz), we investigated their influence on the observed distributions. While shorter recordings can effectively capture cardiovascular events, they may underestimate the variables describing their distribution. Additionally, the tipping point of the Zipf distribution differs between bradycardia and tachycardia events. Comparisons of the distribution of bradycardia and tachycardia events should be conducted using long data recordings. Utilizing devices with lower sampling frequencies may compromise data fidelity. These insights contribute to refining experimental protocols and advancing our understanding of the complex dynamics underlying cardiovascular regulation.

## 1. Introduction

Self-organized criticality (SOC) has emerged as a pivotal concept in understanding the behavior of complex systems across various disciplines, from physics to biology and beyond [1,2,3]. While self-organized criticality has garnered significant attention and sparked considerable research interest since its introduction by Per Bak, Chao Tang, and Kurt Wiesenfeld in the late 1980s, it remains a concept that is not universally accepted or widely known across the scientific community [4,5]. Critics often raise concerns regarding the empirical validation of SOC in real-world systems [5]. Additionally, the absence of clear experimental evidence or predictive capabilities in certain instances has hindered SOC adoption as a mainstream concept in many scientific communities [5]. Despite these challenges, proponents of SOC continue to refine its theoretical frameworks, develop new empirical methodologies, and explore its applications across various disciplines.

Many of the systems in which self-organized criticality has been observed and described pose significant challenges when it comes to experimental manipulation. Astrophysical studies contend with the inaccessibility of celestial bodies. For other systems, this limitation stems from the inherent complexity and scale of these systems. For instance, natural phenomena, such as earthquakes, forest fires, and avalanches, occur over vast spatial and temporal scales, making controlled experimentation impractical or ethically untenable. Similarly, certain human-made systems, such as financial markets or social networks, exhibit behaviors reminiscent of SOC, but attempting to manipulate these systems in a controlled manner may introduce ethical or legal concerns as they often involve human interactions and societal dynamics.

Contrary to many physical or societal systems, biological systems often offer greater opportunities for experimental manipulation. In laboratory settings, researchers can manipulate various biological parameters, such as gene expression, cellular signaling pathways, and environmental conditions, with precision and control. This experimental versatility enables researchers to elucidate the underlying mechanisms of self-organized criticality in biological systems, offering insights into phenomena such as neuronal avalanches in the brain and immune response dynamics [3]. By leveraging experimental manipulation in biological systems, researchers can not only validate theoretical models of self-organized criticality but also explore its implications for health, disease, and ecosystem dynamics.

On the other hand, the study of self-organized criticality in biological systems faces significant limitations due to the inherent biological processes of life maintenance. Factors such as digestion, circadian rhythms, and other physiological fluctuations introduce continuous changes that challenge the stability necessary for efficient studies on criticality. These fluctuations also contribute to the increase in entropy within the system, further complicating the analysis of critical behavior. The cardiovascular system, with its heart, intricate network of blood vessels, and regulatory mechanisms, represents a complex biological system. In recent years, researchers have uncovered compelling evidence suggesting that the cardiovascular dynamics exhibit characteristics consistent with SOC [6,7,8,9,10,11]. One area of investigation involves the study of heart rate variability (HRV), which reflects the fluctuations in the intervals between successive heartbeats [12,13]. Studies have shown that HRV exhibits fractal-like patterns and power-law distributions, indicative of scale invariance, a hallmark of self-organized criticality [13,14]. Additionally, analyses of blood pressure dynamics and vascular reactivity have revealed similar scale-invariant behaviors, suggesting that the cardiovascular system operates near a critical state [15]. Furthermore, critical dynamics of the cardiovascular system have been implicated in pathological conditions like vasovagal syncope [6,7]. Understanding the role of self-organized criticality in normal cardiovascular function and disease states holds promise for developing novel diagnostic and therapeutic strategies.

Besides the common limitations of biological systems in the study of self-organized criticality as discussed before, the study of cardiovascular self-organized criticality faces a specific limitation since cardiovascular self-organized criticality has been observed mainly in the standing position [8]. Long-duration studies of the cardiovascular system in the standing position encounter limitations due to the inherent challenge of sustaining this posture for extended periods. The human body’s physiological mechanisms, such as fatigue and orthostatic intolerance, make it impractical to maintain an upright stance over prolonged durations [16]. Consequently, researchers face constraints in gathering comprehensive data on cardiovascular responses in standing positions, necessitating alternative methodologies or shorter study durations to mitigate these limitations. Zipf’s law was initially described in heart rate variability by [9]. Subsequent studies have corroborated this observation [10,11]. Recently, we proposed evaluating Zipf’s law through the distribution of bradycardia events, as they are associated with vasovagal syncope, a catastrophic event in cardiovascular self-organized criticality [7]. Bradycardia events are abundant and readily recorded, simplifying the assessment of cardiovascular self-organized criticality. The question remains as to whether tachycardia events may exhibit the same distribution as bradycardia events.

Our study aims to offer methodological insights into investigating cardiovascular self-organized criticality through the distribution of both bradycardia and tachycardia events according to Zipf’s law, with a focus on the necessary duration and quality of data recording.

## 2. Material and Methods

### 2.1. Ethical Approval

Patients were provided with a comprehensive explanation of the experimental procedure prior to providing their written informed consent. The experiment, conducted in accordance with the Declaration of Helsinki, Finland, received approval from the Comité Consultatif de Protection des Personnes dans la Recherche Biomédicales des Pays de la Loire (Regional Committee for the Protection of Persons, #00/08, 30 May 2000) in France.

### 2.2. Subjects

The data analyzed in this study were obtained from healthy subjects participating in a larger investigation on vasovagal syncope [17]. In this comprehensive study, one hundred consecutive patients presenting with symptoms suggestive of vasovagal syncope and who provided informed consent underwent a 45 min head-up tilt test or a shorter test if symptoms occurred. The head-up tilt test procedure is outlined below. Patients with a positive head-up tilt test were excluded from the analysis presented in this study. Additionally, patients with a negative head-up tilt test but who exhibited cardiovascular adaptations indicative of orthostatic hypotension or postural orthostatic tachycardia syndrome, those with a history of cardiac rhythm disorders, tobacco users, or individuals on chronic medication were also excluded. The data analyzed herein pertain exclusively to the remaining healthy subjects meeting these criteria.

### 2.3. Head-Up Tilt Test

The head-up tilt test is a valuable tool for clinicians to evaluate a patient’s response to orthostatic stress and to diagnose conditions such as orthostatic hypotension or vasovagal syncope [16]. During the 45 min test, the patient began in a supine position on a tilt table, with baseline vital signs, including heart rate and blood pressure, recorded. The table was then slowly tilted to a 70-degree angle, simulating an upright posture. Throughout the test, the patient’s condition was closely monitored for signs of dizziness, fainting, or changes in vital signs. The test continued for 45 min or until symptoms appeared. If symptoms occurred or the test duration ended, the patient was returned to the supine position and monitored until stable. The head-up tilt table test was conducted in accordance with established recommendations [16]. A detailed procedure description can be found in the comprehensive investigation report [17]. In brief, the test was administered using a motorized inclination table equipped with a platform (AkronA8622, Electro-Medical Equipment, Marietta, GA, USA) within a quiet room featuring subdued lighting and a consistent ambient temperature of 22–24 °C. The patients were instructed to maintain silence but to promptly report any symptoms. They were also instructed to keep as still as possible with their eyes open.

### 2.4. RR-Interval Time Series

The electrocardiogram was continuously recorded throughout the entire 45 min in the head-up position (MAC vu, Marquette, Milwaukee, WI, USA). The sample frequency was set at 500 Hz (AT-MIO-16, 12 bits, Labview 5.1, National Instruments, Austin, TX, USA). R peaks were automatically detected offline using custom-made software based on amplitude and first derivative of the ECG waveform. Any misdetections and ectopic beats were manually corrected by a trained operator who replaced the erroneous values with the preceding ones. Conventional interpolation filtering was not employed, as it can artificially generate cardiovascular bradycardia or tachycardia events. We determined RR-intervals at the resolution of the data acquisition. Beat-by-beat, RR-interval, continuous time series were obtained for each subject over a period of 40 min in the head-up position. The first five minutes following the transition to the head-up tilt were discarded, and only the last 40 min of the test were considered for RR-interval analysis.

### 2.5. Zipf’s Distribution Graphs

The analysis of cardiovascular self-organized criticality was conducted graphically using the distribution of bradycardia events based on Zipf’s law, as previously described [7]. Bradycardia events are routinely observed in heart rate variability recordings, characterized by a consecutive drop in heart rate over several beats [7]. Such events are unexpected, as regulatory mechanisms typically prevent them to maintain heart rate within normal values, as per homeostasis theory. Data regarding the occurrence of these events, categorized by their length in beats, were collected from each RR-interval recording. An RR-interval was deemed distinct from the preceding one at the resolution of RR-interval measurements without imposing any threshold for minimum or maximum differences, as observed in prior studies employing similar methods [6,7]. Consecutive RR-intervals that either increased or decreased were classified as bradycardia or tachycardia events, respectively. They were quantified based on their length in terms of the number of beats. The ranking of events of equal length was established by organizing them according to their frequency of occurrence. The rank of events of the same length was determined by sorting them based on their frequency of occurrence. A decimal log-log plot was generated for each recording, illustrating the frequency of events on the x-axis based on their rank and the length of events on the y-axis in terms of the number of beats.

### 2.6. Position of the Tipping Point

The power law exhibited by the Zipf’s distribution of bradycardia events experiences a breakage approximately at a delay of four-beat intervals, resulting in two distinct straight lines instead of a single one, with a tipping point separating them. The first line pertains to the shorter bradycardia events, while the second corresponds to the longer ones. To enhance the precision of determining this tipping point’s position, the best linear fit was established for each RR-interval recording using multiple linear regressions. We conducted repeated linear regressions by iteratively removing the data point corresponding to the longest events to establish the optimal linear fit for the distribution of short events and, likewise, by iteratively removing the data point corresponding to the shortest events to determine the best linear fit for the distribution of long events. The tipping point was identified as the mean value between the number of data points of the best linear fit for the short and long events.

### 2.7. Variables Determined on the Zipf’s Distribution Graphs

Linear regressions were conducted on each Zipf’s distribution graph based on the previously determined tipping point. At least one linear regression was generated for the short events in each graph, providing regression coefficient and slope values. For long events, an additional linear regression was performed when possible, yielding regression coefficient and slope values as well. A distribution was deemed to fit a straight line when |r| > 0.95 [7]. The maximum length of the events was determined for each graph.

### 2.8. Study on the Influence of Methodology

RR-interval recordings were manipulated to assess the impact of methodological variables on the analysis of self-organized criticality. These manipulations involved altering the duration of the data recording and adjusting the sampling frequency. The original 40 min RR-interval time series were subdivided into shorter recordings of 1 min, 2 min, 5 min, 10 min, 20 min, and 30 min to examine the effect of recording duration. The short recordings always commenced from the beginning of the overall 40 min recording to align with the short-recording durations in usual experimental conditions. Additionally, the individual RR-interval values from the 40 min series were resampled at frequencies of 250 Hz and 100 Hz to investigate the influence of sampling frequency.

### 2.9. Cardiovascular Events

RR-interval recordings were analyzed following the same methodology outlined for the bradycardia events but were extended to include the examination of tachycardia events. Tachycardia events represent a series of consecutive heartbeats characterized by an increase in heart rate. This analysis was conducted on both the original 40 min RR-interval recordings and their shortened versions spanning 1 min, 2 min, 5 min, 10 min, 20 min, and 30 min. The differences between the maximal lengths of bradycardia and tachycardia events were obtained for each duration of data recording.

### 2.10. Statistics

Data were presented as the mean ± standard error of the mean (SEM). Comparisons were conducted among recordings of varying durations as well as between recordings at different sampling frequencies. The normality of the data was assessed using the Kolmogorov–Smirnov test. Comparisons between conditions were made using the Friedman nonparametric test for repeated measurements, followed by Dunn tests when applicable. A mixed-effects model was not employed to handle missing values, as these were not due to technical or experimental issues. Instead, the missing values arose from the intrinsic dynamics of the cardiovascular system under the studied conditions. Each missing value provided meaningful information about these dynamics and could not be statistically disregarded. Statistical significance was considered at a level of *p* < 0.05.

## 3. Results

### 3.1. Subjects

Out of the initial group of one hundred consecutive patients, thirty were excluded because either their interview did not suggest vasovagal syncope or they had a history of heart disease. Additionally, the head-up tilt test identified three patients with orthostatic hypotension and five patients with postural tachycardia syndrome, totaling eight exclusions. Among the remaining 62 patients, 34 exhibited (near) syncope symptoms during the head-up tilt test and were, therefore, excluded. From the remaining 28 patients who tested negative on the tilt test, seven were free from medication, tobacco use, and known chronic diseases. The focus of the present study was on these seven healthy subjects who maintained the head-up position for the entire 45 min duration of the tilt test.

### 3.2. Analysis of the Whole 40 Min Data Recordings

#### 3.2.1. Overall Pattern of Cardiovascular Event Distribution

The overall pattern of cardiovascular event distribution is characterized by two straight lines with a tipping point between them, as previously observed. This pattern is evident in both the distribution of bradycardia events and tachycardia events (Figure 1).

#### 3.2.2. Tipping Point Position

The position of the tipping point varied between the distributions of bradycardia and tachycardia events. In the analysis of the bradycardia event distribution, the tipping point was observed at four-beat intervals (3.9 ± 0.3 beats), whereas for tachycardia events, it occurred at five-beat intervals (4.9 ± 0.2 beats; Figure 1). These specific values will be used in the subsequent studies presented here for all subjects and across all tested recording durations.

#### 3.2.3. Quality of the Straight Line Fit

The regression coefficient of the shortest bradycardia distribution did not meet the threshold value to confirm a straight line in four out of the seven subjects, resulting in a mean coefficient of regression below this threshold for the entire group (0.94 ± 0.01; Figure 1). Conversely, the regression coefficient of the longest bradycardia distribution exceeded the threshold for all seven subjects (0.99 ± 0.00; Figure 1). Additionally, both the shortest and longest tachycardia distributions exhibited regression coefficients above the threshold for all seven subjects (0.97 ± 0.00 and 0.99 ± 0.00, respectively; Figure 1).

### 3.3. Analysis of the Data Recordings According to Their Length

The duration of data recordings significantly influenced the maximum length of cardiovascular events observed in these recordings (Figure 2c, exact *p* values are for bradycardia 40 vs. 1: 0.0001; and 40 vs. 2: 0.032; and for tachycardia 40 vs. 1: 0.0001; 40 vs. 2: 0.0006; 40 vs. 5: 0.0146; and 40 vs. 10: 0.0219). Shorter recordings logically missed the longest cardiovascular events observed in longer-duration recordings. However, this difference was particularly notable for very short-duration recordings of 1 and 2 min lengths but less pronounced for the typical durations of cardiovascular recordings, such as 5 and 10 min lengths, and not significantly different for recordings of at least 20 min (Figure 2c). Interestingly, the length of tachycardia events appeared to be longer than that of bradycardia events, particularly noticeable in recordings lasting at least 20 min (Figure 2c). However, this difference was not significant, since the delta of the longest tachycardia event minus the longest bradycardia event was not different across the tested durations of data recording (1.1 ± 0.7; 1.1 ± 0.5; 1,0 ± 0.5; 0.9 ± 0.5; 3.1 ± 1.0; 3.6 ± 1.2; 3.7 ± 1.3, for 1; 2; 5; 10; 20; 30; and 40 min duration of data recording, respectively; *p* = 0.052). The duration of data recordings did not influence the slope of the distribution of short bradycardia or tachycardia events (Figure 2a). Similarly, the duration of data recordings did not significantly impact the slope of the distribution of long bradycardia or tachycardia events (Figure 2b). However, this latter result should be interpreted cautiously, as there were several instances of missing data, especially in the shortest data recordings. The slope for long bradycardia events could not be determined in five out of seven subjects in the 1 min recordings, four subjects in the 2 min recordings, three subjects in the 5 min recordings, two subjects in the 10 min recordings, and one subject in the 20 and 30 min recordings. It was determinable in all seven subjects in the 40 min recordings. Similarly, the slope for long tachycardia events could not be determined in three subjects in the 1 min recordings and two subjects in the 2 min recordings, but it was determinable in all seven subjects in recordings lasting at least 5 min.

### 3.4. Analysis of the Data Recordings According to the Sampling Frequency of the Electrocardiogram Signal

Only the lowest sampling frequency of 100 Hz affected the distribution of both bradycardia and tachycardia events. The absolute values of the slopes of the distribution of short events and the total number of observed events were smaller in recordings sampled at 100 Hz compared to those sampled at 250 or 500 Hz. The maximum length of observed bradycardia events was also smaller in recordings sampled at 100 Hz compared to those sampled at 250 or 500 Hz (Table 1; exact *p* values for maximum length of bradycardia events at 500 Hz vs. 100 Hz: 0.0222; for the slope of short bradycardia events at 500 Hz vs. 100 Hz: 0.0004; for the total number of observed bradycardia events at 500 Hz vs. 100 Hz: 0.0004; for the slope of short tachycardia events at 500 Hz vs. 100 Hz: 0.0006; for the total number of observed tachycardia events at 500 Hz vs. 100 Hz: 0.0006).

## 4. Discussion

This study elucidates several methodological considerations regarding the technical prerequisites for investigating cardiovascular self-organized criticality via heart rate variability. Research endeavors can encompass evaluations of both bradycardia and tachycardia events. The latter can aid in characterizing the distribution of cardiovascular events to assess self-organized criticality, particularly in instances of brief recordings. However, when utilizing short recordings, it may not be feasible to acquire all variables describing the distribution of cardiovascular events. The use of devices with the lowest sampling frequencies is not advisable.

Since its introduction in the late 1970s, heart rate variability has served as a crucial indicator of autonomic nervous system function [13]. Assessment typically involves both short-term and long-term data recordings, reflecting the two primary methods for obtaining RR-interval recordings: electrocardiograms in laboratory settings or Holter monitors, wearable devices capable of recording electrocardiograms over the course of a day in daily life [13]. Short-term recordings typically span 5 min, a duration determined by the technical requirements of spectral analysis, a key tool for studying heart rate variability [13]. These recordings offer insights into immediate physiological responses to stimuli or activities. Conversely, long-term Holter recordings reveal trends and fluctuations over time. Our study focuses on short-term cardiovascular recordings, suitable for investigating the effects of experimental manipulation or environmental adaptation, such as the standing position. Extending the duration to 40 min provides insights into the significance of cardiovascular event distribution observed in more typical-duration recordings. Despite the consensus on the duration of short-term cardiovascular recordings, there are ongoing efforts to further reduce this duration with very short-term recordings [18]. The objective of these efforts is to enhance the information accessible during cardiovascular monitoring by computing and displaying derived variables online from the raw signal. This study also incorporates very short-term recordings to compare the quality of information against more typical short-term cardiovascular recordings and extended ones. The findings of this study demonstrate that, as anticipated, longer recordings yield better results. However, it also indicates that regular-duration cardiovascular monitoring may still provide reliable information about cardiovascular self-organized criticality, albeit requiring cautious interpretation.

Most studies on cardiovascular self-organized criticality have been conducted using Holter monitors, which provide readily available recordings, including those accessible through internet databases [9,10,11]. However, the primary aim of these initial studies was to seek evidence of self-organized criticality within these recordings. Now is the time to delve into understanding how this self-organized criticality influences cardiovascular function and blood pressure regulation. This understanding could be enhanced by short-term data recordings during experimental manipulations of the cardiovascular system to test its self-organized criticality [8]. Sampling frequency requirements for electrocardiogram signals are well-defined due to the importance and widespread use of heart rate variability [13,19]. These requirements are outlined in consensus statements, which suggest that an electrocardiogram should be sampled at a frequency of more than 250 Hz, with a higher frequency recommended for instances of elevated heart rates, such as those observed during exercise. However, a sampling frequency exceeding 1000 Hz is not advised, as it does not enhance heart rate variability analysis [13]. These recommendations are primarily based on experiments that evaluated the impact of sampling frequency on heart rate variability, as assessed through spectral analysis. Interestingly, these recommendations have been found to apply to other heart rate variability tools as well [20]. Our study demonstrates that these recommendations also hold true for the study of self-organized criticality, despite employing a completely different approach.

Cardiovascular self-organized criticality has been observed and studied by examining the statistical distribution of bradycardia events, as they are associated with vasovagal syncope [6,7]. During vasovagal syncope, the cardiovascular pattern typically involves both bradycardia and hypotension [16]. Long bradycardia events are unexpected in cardiovascular dynamics due to regulatory mechanisms. However, tachycardia events, characterized by an increase in heart rate over several heartbeats, are also unexpected for the same reason. Our study demonstrates that cardiovascular self-organized criticality can be assessed through the distribution of both bradycardia and tachycardia events. Additionally, our findings reveal a higher occurrence of tachycardia events compared to bradycardia events. However, the tipping point of the tachycardia event distribution occurs for longer events than in the case of bradycardia events, which limits the description of this distribution for tachycardia events. The higher prevalence of tachycardia compared to bradycardia can be quantified, providing valuable variables for studies on cardiovascular self-organized criticality. However, this trend requires further investigation. Our results suggest that exploring this disparity requires longer data recordings, as the difference seems to increase with recording duration. Additionally, a larger population sample is needed beyond the small group studied in this research.

The primary hallmark of self-organized criticality is the presence of power laws evident in the distribution of catastrophic events within the dynamic of the studied system [1,3]. Investigating this distribution entails observing the system for a sufficiently lengthy duration to capture numerous such catastrophic events. However, conducting such prolonged observations is not always technically feasible. Furthermore, the continuous changes inherent in biological systems present a challenge for long data collection, as previously mentioned. Self-organized criticality can also be studied through the power law exhibited by the system’s dynamics, independent of the occurrence of catastrophic events [7]. The dynamics of the cardiovascular system provide an opportunity to explore its self-organized criticality through the distribution of catastrophic events as well as independently of them. The latter perspective requires shorter data recordings than the former. However, the question remains: How short can this data collection be while still providing meaningful insights? This was one of the primary objectives of our study. We demonstrated that certain variables describing the self-organized power law could be determined even with relatively short-term data recordings lasting only a few minutes, although interpretation should be approached with caution. The cardiovascular system also provides an intriguing perspective on self-organized criticality, as the characteristics of the power law associated with self-organized criticality change in response to environmental challenges such as posture [8]. Further studies are needed to address the puzzle of a universal theory that applies in the standing position but not in the supine position. The cardiovascular system offers an opportunity to investigate whether physiological systems naturally operate at criticality or are poised close to criticality, while physiological regulations prevent excessively high criticality. The ability to easily characterize self-organized power laws using short data recordings offers the potential for identifying diagnostic markers. Information obtained from very short-term data recordings could also be valuable for obtaining useful measurements during online patient monitoring.

The primary challenge in studying cardiovascular self-organized criticality lies in the relatively narrow range of approximately two orders of magnitude of cardiovascular event lengths. By comparison, earthquake magnitudes, often used as a point of reference, encompass a range of approximately nine orders [1]. However, various other dynamical systems exhibit self-organized criticality with similarly limited distribution ranges, such as coastal fractality, sediment deposition, pulsar glitches, the game of life, and the punctuated equilibrium model of evolution [1]. This restricted range may be attributed to the constant adaptation of biological systems to maintain life and adhere to circadian rhythms. While a narrow range could potentially limit definitive conclusions regarding the presence of self-organized criticality in a dynamical system, the cardiovascular system presents additional compelling evidence. Power laws, such as Zipf’s law employed in this study, have consistently been described in cardiovascular dynamics using various methodologies [7,9,10,11]. Another pertinent power law, the Gutenberg–Richter law, directly correlates with the distribution of catastrophic cardiovascular events, such as vasovagal syncope or common faints, affecting approximately one in three individuals in the general population [6]. The 1/f pattern observed in cardiovascular time series, dating back to the 1980s, is characteristic of self-organized system dynamics and further supports the presence of self-organized criticality in the cardiovascular system [14]. Additionally, non-equilibrium phase transitions, observed during sleep, strenuous exercise, and positional changes, are indicative of self-organized system behavior in heart rate variability [15,21].

The narrow range of lengths in cardiovascular events poses a challenge in conclusively demonstrating cardiovascular self-organized criticality. However, this limitation does not impede the study of cardiovascular self-organized criticality in response to environmental conditions or challenges. Fortrat and Ravé have shown that even brief recordings as short as 5 min can reveal the significant impact of the standing position on cardiovascular self-organized criticality [8]. While longer recordings offer more comprehensive insights, shorter ones can still yield valuable observations, albeit with less optimal descriptions of cardiovascular events. These short recordings present an opportunity to gain a deeper understanding of cardiovascular self-organized criticality in relation to environmental adaptations and challenges. Moreover, they offer a chance to explore the potential applications of self-organization in monitoring and diagnosis.

The current study focuses on utilizing cardiovascular events to evaluate self-organized criticality. These events serve not only to assess criticality but also to explore other properties of the cardiovascular system [12,22]. Notably, the baroreflex plays a pivotal role in regulating blood pressure within its normal range during acute cardiovascular challenges such as position shifts. Assessing the sensitivity of this reflex involves examining cardiovascular events known as baroreflex sequences [22]. While distinct from the bradycardia and tachycardia events analyzed in our study, baroreflex sequences represent another aspect of beat-by-beat heart rate and blood pressure dynamics. Recently, bradycardia events, referred to as heart rate decelerations, have emerged as potential risk markers for sudden death in post-infarction cardiovascular monitoring [23]. Intriguingly, these decelerations are observed to increase in patients experiencing vasovagal syncope, a catastrophic event within the realm of cardiovascular self-organized criticality [24]. Gañán-Calvo and Fajardo-López refined the concept of cardiovascular sequences to propose a graphical analysis of heart rate variability, revealing two universal arrhythmic patterns as specific health signatures: one reflecting cardiac adaptability and the other indicating cardiac–respiratory rate tuning [12].

## 5. Study Limitations

This study was conducted with participants in a passive head-up tilt position, and the findings might vary if other positions, such as active standing, were used. Sustaining an active standing position for the duration of 40 min, as performed in this study, is almost impractical. The participants were recruited based on symptoms indicative of vasovagal syncope, potentially setting them apart from the general population. The participant selection also introduced a survival bias by excluding a significant portion of the initial population, resulting in a small sample size. These factors preclude drawing conclusions about the general population. However, this study’s objective was not to offer insights into the general population but rather to present methodological considerations. Examining these participants provided an opportunity to analyze long-duration data recordings in the head-up position, offering valuable insights into the methodology required to study cardiovascular self-organized criticality. It is worth noting the high incidence of vasovagal symptoms in the general population, with one in three people experiencing such symptoms. This prevalence underscores its status as a widespread phenomenon that could occur in anyone. Finally, this study’s primary limitation lies in its exclusive focus on methodological aspects, neglecting insights into the criticality and self-organizing dynamics of heart rate variability. While several arguments support the notion, the question of cardiovascular self-organized criticality remains unresolved. One such argument involves the observation of power laws within heart rate variability. However, the mere presence of a statistical distribution following a linear trend in a log-log graph is insufficient to establish self-organized criticality. Clauset et al. have proposed more robust analytical methods beyond simple linear regression [25]. While our methodological investigation offers some indication, it underscores the necessity of prolonged data collection for applying these advanced tools to arrive at a more conclusive answer.

## 6. Conclusions

In conclusion, our study offers significant methodological insights into exploring cardiovascular self-organized criticality (SOC) through heart rate variability analysis. We have shown that even short recordings can sufficiently capture cardiovascular events, albeit with less-than-optimal distribution descriptions. This highlights the importance of tailoring recording duration to match research objectives and dynamics of interest. A nuanced comprehension of methodological considerations in investigating cardiovascular SOC will enhance the development of more resilient analytical strategies and aid in translating research discoveries into clinical applications. Ultimately, this advancement holds promise for enhancing the management of cardiovascular disorders.

## Figures and Tables

**Figure 1 entropy-26-00496-f001:**
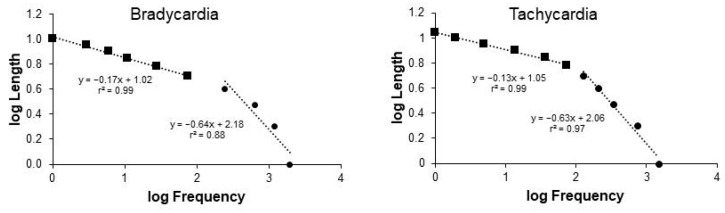
**Zipf’s distribution of cardiovascular events.** The left panel displays an example of Zipf’s distribution graph for bradycardia events observed in one of the subjects, while the right panel shows the corresponding graph for tachycardia events. Cardiovascular sequences are sorted and counted based on their frequency of occurrence and length in terms of the number of beats. The x-axis represents frequency, while the y-axis represents event length in base-10 logarithms. Distribution graphs are revisited following analysis to determine their tipping point and subsequent linear regression of short and long events (black circles and squares, respectively).

**Figure 2 entropy-26-00496-f002:**
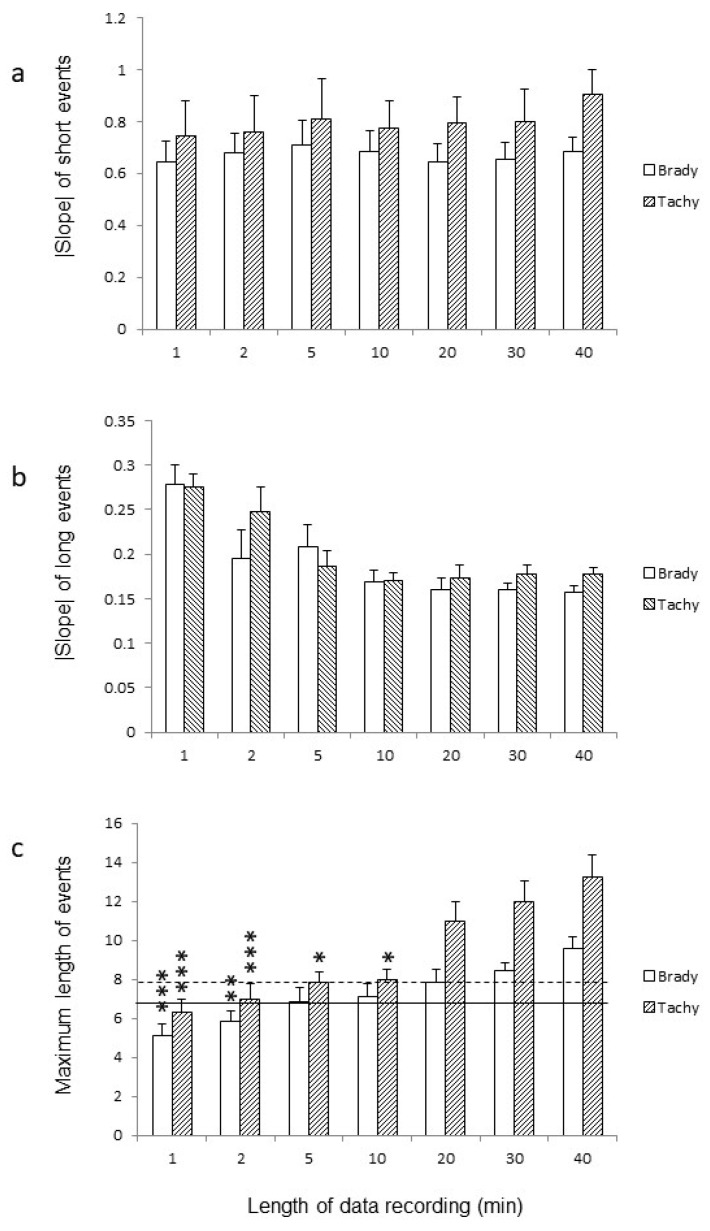
**Variables determined on the Zipf’s distribution graphs.** The figure illustrates the slopes of short and long cardiovascular events as well as the maximum length of these events determined from Zipf’s distribution graphs, plotted against the length of the data recordings (x-axis). The upper panel a displays the slope of short cardiovascular events, while the middle panel b depicts the slope of long cardiovascular events. The lower panel c represents the maximum length of the cardiovascular events. Statistical significance is denoted by * *p* < 0.05, ** *p* < 0.01, and *** *p* < 0.001. The solid and broken horizontal lines indicate the minimum length of bradycardia and tachycardia events, respectively, required to determine the slope of long cardiovascular events.

**Table 1 entropy-26-00496-t001:** Influence of sampling frequency of the electrocardiogram.

Bradycardia	100 Hz	250 Hz	500 Hz
Max length	7.6 ± 0.4 *	8.9 ± 0.4	9.6 ± 0.6
|Slope| of short events	0.56 ± 0.04 ***	0.65 ± 0.05	0.68 ± 0.06
|Slope| of long events	0.14 ± 0.02	0.15 ± 0.01	0.16 ± 0.01
*n*	1358 ± 77 ***	1551 ± 92	1608 ± 91
Tachycardia			
Max length	12.3 ± 1.3	12.4 ± 1.1	12.6 ± 1.0
|Slope| of short events	0.752 ± 0.096 ***	0.859 ± 0.095	0.884 ± 0.090
|Slope| of long events	0.181 ± 0.012	0.171 ± 0.010	0.172 ± 0.009
*n*	1377 ± 99 ***	1543 ± 83	1576 ± 81

The sampling frequencies of 100 Hz, 250 Hz, and 500 Hz were compared for their efficacy in determining descriptive variables of bradycardia and tachycardia event distribution. These variables included the maximum length of observed events (Max length), the slope of short event distribution, the slope of long event distribution, and the total number of observed events regardless of length (*n*). Statistical significance was indicated by * *p* < 0.05 and *** *p* < 0.001 compared to 250 Hz and 500 Hz, determined using the Friedman test followed by Dunn’s test when appropriate.

## Data Availability

The data presented in this study are available on reasonable request from the corresponding author.

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
