# Peer review of "Purported Self-Organized Criticality of the Cardiovascular Function: Methodological Considerations for Zipf’s Law Analysis"

_entropy, 2024, doi:10.3390/e26060496_

Round 1

Reviewer 1 Report

Comments and Suggestions for Authors

In this manuscript, the author aims at providing a list of methodological considerations to apply the theory of self-organized criticality to the field of cardiology and especially to heart rate variability. Here, the focus is put on Zipf’s law and, in particular, the impact of recording durations and sampling frequencies on the analysis of the Zipf diagrams for both bradycardia and tachycardia sequences. Even though the number of subjects is small (7), the manuscript provides promising information regarding the requirements to properly perform this type of analysis. While some aspects require further work (see detailed description), this manuscript fills a gap in the literature, as self-organized criticality is starting to show its interest in the cardiovascular field (e.g., to evaluate the susceptibility to vasovagal syncope). It is indeed crucial that some guidelines are provided to the authors desiring to apply such techniques in their research. This will uniformize the way these parameters are used and make future comparisons / reviews easier.

Abstract

It would be interesting to provide guidelines in the abstract, e.g., lowest sampling frequency / recording duration depending on the accuracy required. Also, it may be interesting to know the impact of a suboptimal sampling frequency / recording duration. Is it an under/overestimation of the parameters or interest? Or simply more noise around the exact value?

Introduction

The introduction clearly contextualizes the topic and states the interest of applying self-organized criticality to the study of heart rate variability.

The introduction (and maybe even the title) should make it clear that this manuscript is focusing in particular on Zipf’s law.

Material and Methods

Paragraph 2.3.: Since many parameters are known to influence heart rate variability, it seems mandatory to give more details regarding the experimental setup. Subjects asked to remain silent? Keep eyes open? Not move at all, including fidgeting? No sound in the room? Illumination level? Operator checking that no muscle pump was used? Etc.

Paragraph 2.4.: How were corrected the ectopic beats? By interpolation?

Paragraph 2.4.: Given the importance of the R peaks detection (underlined by the effect of sampling frequency on your features of interest), I would suggest adding more information regarding their detection within the “custom-made software” that was used. Was it an implementation of the Pan-Tompkins algorithm?

Paragraph 2.4.: I suppose that the 40 minutes considered for analyses do not include the first 5 minutes following the transition to head-up tilt. This should be explicitly mentioned: “The first five minutes following the transition to head-up tilt were discarded, and only the last 40 minutes of the test were considered for RR-interval analyses.”.

Paragraph 2.5.: Based on Figure 1, it looks like you are using a log10. It would be interesting to write it in the methodology, especially to avoid confusion with other authors potentially using a natural logarithm instead.

Paragraph 2.6.: It is not easy to understand how the tipping point was chosen. Are you starting from a set of a few points and then iteratively adding new points and checking the R² until it stops decreasing?

Paragraph 2.7.: By construction, is the tipping point systematically between two lengths on the loglog plot? Or does it correspond to a specific length on this loglog plot? If so, is the tipping point included in the short- or in the long-sequences of bradycardia? Or none of them?

Paragraph 2.8.: Which part of the signal was chosen for the shorter recordings of 1, 2, 5, 10, 20, and 30 minutes? Is it always the beginning (or the end) of the overall record (after the 5 minutes of transition)? If so, this may introduce a bias. The best may be to take them at random.

Paragraph 2.9.: This comes a bit late. I would expect to see that tachycardia will be studied in the introduction as well.

Paragraph 2.10.: I am not sure to understand why the Friedmann test was followed by a Kruskal-Wallis test. Why not using a Friedman test followed by paired non-parametric comparisons (Wilcoxon signed-rank tests, for instance)? Also, how did you deal with the missing data with a Friedmann test? A linear mixed effects model may be better in such cases.

Results

Paragraph 3.2.: Figure 1 and Figure 2 could be combined, as they are the same figure, except for the visual display of the short- and long-term sequences of bradycardia/tachycardia.

Paragraph 3.5.: Do you confirm that, based on the mean position of the tipping point, you then performed the short-term regression on the first 4 point for bradycardia (5 points for tachycardia), and the long-term regression on the remaining points? If so, maybe this should be clear in the Material and Methods: first you define the mean position of the tipping point and then you use the same one for all the subjects and for all the recording durations / sampling frequencies. If not, then I misunderstood, which means that it may also be explained a bit more clearly.

Line 231: What do statistical tests say? Are the longest tachycardia sequences indeed longer than the longest bradycardia sequences (at least in the 40-minute test)?

Line 233/234: The reference should be “Figure 3a” (even though so far there is no letter on the figure itself).

Figure 3: Add letters to the subplots, so that it easier to reference them in the text. I am not sure that it is necessary to indicate the four levels of statistical significance (however, the p values can be mentioned in the text). Also, it appears strange to me that p < 0.0001 is even possible for only 7 subjects. On the y axis of Figure 3a, I would indicate “Slope of short sequences” instead of simply “Slope”. Same thing for Figure 3b: “Slope of long sequences”, instead of simply “Slope”. Finally, the slopes that you report in this figure are positive, while the ones shown on Figure 2 are negative. I would suggest to either flip horizontally the data on Figure 2 (reverse the x axis and have positive slopes) or to report negative slopes in the figures/tables.

Paragraph 3.6.: Could you please report the same information for the tachycardia sequences?

Table 1: It looks like you had enough space to write “Slope of short sequences” instead of “a short” (same thing for long sequences). It brings useless confusion to add the variable ‘a’ here. Again, it does not seem relevant to report the different levels of statistical significance. However, the actual p values can be indicated in the text.

It would be interesting to evaluate the impact of where you take the shorter records within the 40 minutes of record. Is it the same if it is at the beginning or at the end of the overall record? If it is different (which may also be seen by looking at RR or blood pressure), this may be another reason why different results are found for different lengths of records. This aspect should at least be discussed.

Discussion

Line 328: I would rather say “Long bradychardia sequences are unexpected”, because there will always be bradychardia sequences in a signal.

Line 334: It is not clear how the higher prevalence of tachycardia sequences compared to bradycardia sequences could facilitate studies on cardiovascular self-organized criticality.

Line 340: Your study indeed shows that the bradycardia/tachycardia disparity in terms of longest sequences increases with time of recording. However, it does not show that the relative number of bradycardia vs. tachycardia is changing with the length of the record (even though it would be interesting to look at this).

Line 360-362: It would be nice to add a reference for this (Fortrat & Ravé, 2023?).

Line 370: Not sure to understand “spanning only about two units”. Do you mean “ranging approximately between 1 and 10”? Or are you talking about the log10 of this interval (thus ranging between 0 and 1)?

The organization of the Discussion is a bit challenging, with many elements already described in the Introduction. In this section, I would recommend focusing more on the results from this study and their interpretation.

Please include a paragraph dedicated to the limitations of this study. This could include, among other aspects: 1) the fact that this study was conducted in a passive 70° head-up tilt position and that the results may be slightly different for other positions, such as active standing; 2) the fact that the subjects were recruited because of symptoms suggestive of vasovagal syncope, which may make them different from the general population; 3) the possibility of a survival bias, since the only subjects considered are the ones who managed to reach the end of the test; 4) the low number of subjects.

Conclusions

Same remark as for the abstract.

Comments on the Quality of English Language

Here is a (non-exhaustive) list of the few typos that I noticed while reading the manuscript:

Line 61: “On the other and” -> “On the other hand”.

Line 82: “the study cardiovascular self-organized criticality” -> “the study of cardiovascular self-organized criticality”.

Line 121: “continues” -> “continued”.

Line 193: Remove this line.

Line 211: “Zipf’’s law” -> ”Zipf’s law”.

Line 258: “slope” -> “slopes”.

Line 418: “We’ve” -> “We have”.

Reviewer 2 Report

Comments and Suggestions for Authors

Review: Self-Organized Criticality of the Cardiovascular Function: Methodological Considerations by J-O Fortrat.

The author presents a methodological paper in which he clearly defines necessary outlined for Data acquisition and analysis using measures of self-organized criticality, here, Zipf’s distribution for bradycardia and tachycardia sequences derived from short and long data sequences. A major purpose of this paper is to help future projects which use these types of analyses for a proper data acquisition.

To study the impact of recording length and sample frequency on Zipf’s distribution analyses the author used real data from a study in patients from which a small cohort of healthy humans were selected. Recording length was varied from 1 up to 40 minutes and sampling frequency to detect RR-intervals was varied between 100 Hz and 500 Hz. For all conditions, Zipf’s distribution was calculated and the key parameters i.e. sequence length frequency and the respective slopes were calculated and compared.

The major results of this study are that sampling frequency should at least be 250 Hz or higher and even if short recording periods still allow a calculation of Zipf’s distribution, a longer recording period results in more reliable analyses.

The applied methods are state of the art and the selection of individuals under test appears reasonable. The manuscript clearly describes the experimental approach, the data generation and analysis and provides adequate presentation.

Reviewer 3 Report

Comments and Suggestions for Authors

This paper focuses on a fairly simple statistical method for analysing heart rate variability (HRV), based on binning of beat frequency and number of beats for each frequency. The author uses the classical Zip’s representation (I would avoid the use of “law” when no such Zip’s power law emerges in the analysis along the whole range of variables). However, the author provides a very poor description of the methodology (e.g. the binning criteria are not detailed). In fact, when the author mentions the term "sequence", any reader with some expertise would immediately think of "temporal signatures" of the beat-to-beat signal of the heart (see e.g. Ganan-Calvo and Fajardo-Lopez 2016, Sci. Rep. 6: 21749). This leads to a fundamental confusion that needs to be clarified in the relevant literature, not only in this work. I therefore invite the author to clarify this issue, firstly by indicating the maximum variability between beats within a bin allowed in the calculations, as I understand that the author does not consider specific temporal variability patterns or signatures, and secondly by avoiding the use of "sequence" in the context of the author's methodology, as this should be used to refer to temporal beat sequences with their specific patterns (e.g. arrhythmia patterns).

I have one last reservation, which relates to the fact that the term "entropy" does not appear once in the text. In order for this manuscript to be published in Entropy, I think it would be necessary to make at least a tangential mention or reference to the connection between the author's conclusions and the entropy of the system analysed, or something similar.

Reviewer 4 Report

Comments and Suggestions for Authors

In this manuscript, the author builds on his previous publications on signatures of self-organized criticality (SOC) in cardiac data. The features of interest are sequences where the heart rate drops over N consecutive beats (bradycardia) or increases over N consecutive beats (tachycardia). Over a long duration recording of tens of minutes, many such sequences can be observed. When the bradycardia sequence frequency of occurrence is plotted against the duration of the sequence in log-log coordinates, then they observe something like a power law. This is taken to be evidence of SOC. In the present work, the focus is on how recordings of different durations, from 1 min up to 40 mins, will affect the quality of the power laws. This is therefore more of a methodological study rather than one concerned with basic results.

I think the topic of SOC is very interesting and it has received much research attention in recent years within biology, physiology, neuroscience and even sociology. This work therefore might be of interest to a broad section of readers. Using the tools of complexity theory should also be encouraged and I am glad to see this in the present manuscript.

That said, I do have some very serious concerns about how the data are being processed and interpreted.

I want to advise the author that showing a plot that appears to be a power law is not sufficient to demonstrate that the data are critical or SOC. There are several reasons for this.

One of the most common is that the data might actually be better fit by some long-tailed distribution that is not a power law, but that looks similar to a power law. For example, a lognormal distribution in the tail may look by eye to be very similar to a power law. Rigorous statistical tests need to be applied to demonstrate that a power law is the most likely distribution. The least squares linear fit with a high correlation coefficient is not considered enough in present research on criticality. For a glimpse at some of the methods that are now needed, please take a look at Clauset, Shalizi and Newman (2009), Alstott, Bullmore and Plenz (2014) or Marshall, Timme, Bennett et al., (2016). These papers and the accompanying toolboxes describe how you can test your distribution to see if a power law is the most likely fit. If it is not, then any discussion of SOC or criticality is off the table.

A second reason why a plot like that shown in figure 2 is not enough is that even a certified power law does not necessarily indicate criticality or SOC. This is because there are numerous non-critical (unrelated to phase transitions) mechanisms that can produce power laws. For example, exponential growth followed by an exponential truncation time can generate a power law, but this is not caused by a critical phase transition (see Reed and Hughes, 2002, Physical Review E). There is also successive fractionation and other mechanisms (see Mitzenmacher, 2004, Internet Mathematics). Briefly, to show a power law caused by a second order phase transition, you should have two phases, a control parameter that can switch the system from one phase to another, and more than one power law (for example one for size of cascades and another for their duration). While you may not be able to get cascade sizes in your system, you could explore other ways of looking for scaling in your data. This could include detrended fluctuation analysis (DFA) which can detect long range temporal correlations (LRTCs). A very nice article describing these and how to calculate them is (Hardstone, Poil, et al., 2012, Frontiers). They even have a software package that lets you calculate these things (contact Klaus Linkenkaer-Hansen for it, as the link on the paper is currently broken). You should calculate the Hurst exponent, H, and see if it is over 0.5, which is what you get for a random walk. If it is not larger than that, then your case for criticality becomes significantly weaker.

In general, many of these issues have already been addressed in the neuroscience literature where there has been considerable discussion about whether the data are critical or not. For an overview, see (Beggs, 2022, Frontiers). There, several of these criticisms and how they can be met are reviewed in the field of neuroscience.

As for the main topic of the current paper: using shorter data segments will affect the quality of the putative power laws you obtain. Yes, this is what I would expect, and this type of issue has been studied often in neuroscience where people have used different time bins or sampling rates and obtained different critical exponents. So I think you are on the right track, but you need to be sure you have authentic power laws first. For some discussion of how data subsampling affects power laws, you can go to Levina and Priesemann, 2017, or Levina, Priesemann and Zierenberg, 2022.

Overall, I do not want to discourage this type of research; on the contrary I think it is very interesting and worthwhile. I just want to be sure you are aware of the potential pitfalls in this area. I have traveled this path for the last 20 years in neuroscience and I would not want you to repeat my mistakes! I am glad to have gone through this, as I learned a lot. I am hoping to pass this along to you with this review. I wish you the best in your work, and I would be happy to review a revised version.

Sincerely.

Round 2

Reviewer 1 Report

Comments and Suggestions for Authors

Thank you for updating your manuscript according to my previous comments. However, I still have a few remarks:

Line 263: You mention “Figure 2c”, while there is no letter to actually define the subfigures of Figure 2.

Line 281-284: This seems to contradict the observation made on lines 268-270. Please add a p-value to evaluate if this may simply be the consequence of a low statistical power, rather than the absence of an actual difference. Also, please mention in the parentheses that the reported values correspond to the delta “longest tachycardia event - longest bradycardia event”. The whole sentence is a bit unclear so far.

Figure 2: Even though the author mentions that Figure 2 (previously Figure 3) was corrected. It does not appear to be so. In particular, please add the letters ‘a’, ‘b’, ‘c’, to describe each subfigure. Also, please mention what do the error bars represent. On the y axis of Figure 2a, I would indicate “Slope of short sequences”. Same thing for Figure 2b: “Slope of long sequences”. Finally, the slopes that you report in this figure are positive, while the ones shown in Figure 1 and Table 1 are negative.

For Figure 2 and Table 1, I find it quite unusual that the p-values are reported in the caption, rather than in the main text of the manuscript.

Line 299: The slopes are different only for the short events, not the long events. Also, because we talk about a negative number, I would rather mention “the absolute slopes” or “steepness of the slopes”. Please also mention the results related to tachycardia events in this paragraph.

Line 383-384: “the disparity increases with recording duration” does not seem in line with the observation made on lines 281-284. Depending on the p-value, you could talk about a trend, instead.

Line 413: “relatively narrow range of about two units of cardiovascular event lengths”. Since we talk about log-transformed data, wouldn’t it better to talk about “two orders of magnitude of cardiovascular event lengths”?

Line 470-472: Seems to repeat what is already said a few lines before (463-465).

Reviewer 3 Report

Comments and Suggestions for Authors

The author has made considerable efforts to improve the quality of his work along the lines suggested by the referees. As far as my own review is concerned, I will not attempt to respond to the author's response to my central comment, as any discussion in this regard would be irrelevant to the publication and potential quality of this paper.

It is my recommendation that the author reconsider his reluctance to cite Ganan-Calvo and Fajardo-Lopez (2016), as referenced in my previous report. The observation and core claim presented in the present manuscript regarding the scale invariance exhibited in HRV aligns with the core claims made by Ganan-Calvo and Fajardo-Lopez in their work. The aforementioned authors demonstrated that specific sequences of HRV are not limited to bradycardia and tachycardia, but also occur at any average local frequency, a strong demonstration of scale invariance. This is why a central methodology in the work by Ganan-Calvo and Fajardo-Lopez is the normalization of the HRV sequence pattern with the local average cardiac frequency. For example, taking sequences of five beats and normalizing the R-R distance with the average among the five subsequent beats reveals normalized patterns that can be found everywhere along the Holter record. In contrast, Fortrat employs a distinct methodology, utilizing longer records and investigating the universality of specific statistical signatures (utilizing Zip's analysis) in both bradycardia and tachycardia. Nevertheless, the fundamental assertion remains consistent with that of Ganan-Calvo and Fajardo-Lopez.

The failure to cite the work by Ganan-Calvo and Fajardo-Lopez (2016) despite the clear relationship between the present manuscript and that work represents a significant shortcoming. I am confident that the author will both comprehend and implement the methodology proposed by Ganan-Calvo and Fajardo-Lopez in his future analyses, should he have the opportunity to do so. This methodology is the optimal approach for identifying universal, scale-invariant HRV sequences which are potentially indicative of a wide range of cardiovascular conditions. It is my intention here to emphasize the use of the term "sequences" as in the work of Ganan-Calvo & Fajardo-Lopez, in contrast to the more general term "events" as it is now used in Fortrat's paper. This is in line with the new and, in my view, correct terminology assumed by the author to distinguish the methodologies of both authors' groups.

If the author agrees to include a brief discussion around the above in the introduction and methodology, I will not have any further objection to the publication of this nice and interesting work.

Reviewer 4 Report

Comments and Suggestions for Authors

First, I am sorry for the delay in getting my comments back to you on re-review.

Second, I think you have improved the manuscript by inserting a paragraph that described the perspective you are taking toward the study. You are primarily concerned with methodological issues regarding time bin sizes and how those could affect the results of such a study. You are not primarily concerned with establishing that the data do in fact demonstrate self-organized criticality. I think this is a good step and I am glad to see you have taken it.

However, your title still contains the term "self-organized criticality," suggesting that readers will assume that you have taken the position that SOC really does exist in your data. I think this is problematic and potentially misleading. You could amend this by inserting the qualifying word "purported" before the self-organized criticality. Addressing this issue up front and directly will lead to less confusion among readers.

The alternative is to actually do rigorous tests, like those outlined in the Clauset paper, and either establish or refute the claims of SOC in these cardiac data. We had to go through this in neuroscience regarding the issue of neuronal avalanches and it was not easy, but it is mostly a settled issue now. It took time and effort on the part of several labs, but almost nobody now questions whether neuronal data are best fit by power laws. The critiques are now of a different sort, but the field is no longer worried about this power law issue. If history repeats itself, I would expect that you would face many problems if you do not address this issue squarely and decisively soon.

Author Response

We sincerely thank the reviewer for its careful reading and insightful comments on the manuscript. Your feedback has been invaluable in improving the quality of our work.

>You could amend this by inserting the qualifying word "purported" before the self-organized criticality. Addressing this issue up front and directly will lead to less confusion among readers.

We changed the title for: Purported self-organized criticality of the cardiovascular function: methodological considerations

We hope to have the chance in a near future to convince the reviewer that the context of neuroscience and neuronal avalanche is very different from this of a mechanically beating heart. Self-organized criticality story of the latter is rich. It includes numerous steps since the description of cardiovascular catastrophe experienced or witnessed by anybody and called vasovagal syncope during the antiquity.